# LDINet: Latent Decomposition-Interpolation for Single Image Fast-moving Objects Deblatting

## Abstract

The image of fast-moving objects (FMOs) usually contains a blur stripe indicating the blurred object that is mixed with the background. To deblur the stripe and separate the object from the background in the single image, in this work we propose a novel Latent Decomposition-Interpolation Network (LDINet) to generate the appearances and shapes of the objects. In particular, under the assumption that motion blur is an accumulation of the appearance of the object over exposure time and the long blur can be decomposed into several shorter blur parts, the blurry input is first encoded into latent feature maps and then an efficient Decomposition-Interpolation Module (DIM) is introduced to break down the feature maps into discrete time indexed parts corresponding to different small blurs. And the target latent frames are further interpolated according to the provided time indexes with affine transformations, where the feature maps are categorized into the scalar-like and gradient-like parts to effectively capture the intrinsic properties of features warping in the interpolation. Finally, the sharp and clear images are rendered with a decoder. In addition, based on the generated images by LDINet, a Refining Conditional Deblatting (RCD) approach is presented to use post-image-to-image techniques to further enhance the fidelity of the textures and the accuracy of the masks. Extensive experiments are conducted and have shown that the proposed methods achieves superior performances compared to the existing competing methods.

Motion deblurring is a special case of the deblurring task that aims to perform high-quality image restoration from a blurred one caused by the possible moving of the object or the camera. Conventional methods Kupyn et al. (2018; 2019); Wieschollek et al. (2017); Sim & Kim (2019) for motion deblurring mostly adopt a setting that recovers a single clear and sharp image in motion. Recently, some works Jin et al. (2018); Purohit et al. (2019); Xu et al. (2021); Zhong et al. (2022); Rozumnyi et al. (2021a); Zhong et al. (2023) further focus on the finer structures of the blur and learn to generate a sequence of clear images of the object in a chronological order, which is known as sequence from blur or single image temporal super-resolution task.

In this work, we focus on a special case of the sequence from blur task, i.e., deblatting (deblurring and matting) of fast-moving objects (FMOs) Kotera et al. (2019). FMOs, first defined by Rozumnyi et al. (2017), are moving objects that move over a distance greater than their size within the exposure time of the camera in the scene. As a result, the blurry portion of a FMO becomes a stripe due to the long-distance moving, which makes it hard to distinguish the object's appearance. Kotera et al. (2019) have further formulated this deblatting problem to accomplish two goals, i.e., image deblurring which generates a sequence of clear and sharp images from the blurry input, and image matting which separates the object in the scene from the background.

Formally, given a pair of pictures as input, including one background picture and one picture with the blurred fast-moving object, our goal is to generate a sequence of sharp appearances and masks for deblurring and matting according to the given time indexes. From the physical perspective, the formation process of the motion blurred input can be formulated as a temporal integral of the underlying consecutive sharp sub-images over a short exposure time. Since solving the integral is difficult, several approximations have been proposed to simplify the formation model. Kotera & Šroubek (2018) approximate the blurring and matting formation model by convolving a fixed moving

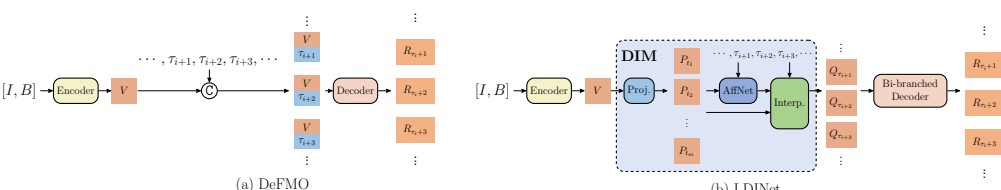

(a) DeFMO          (b) LDINet

Figure 1: **The pipelines of DeFMO and LDINet.** (a) Given the blurred input $I$ and background $B$, DeFMO encodes the inputs into a latent embedding $V$ and augments it with the time index $\tau \in \{\cdots, \tau_{i+1}, \tau_{i+2}, \tau_{i+3}, \cdots\}$ for further rendering the output $R_\tau$ by decoder. (b) Differently, LDINet decomposes the latent embedding $V$ into $m$ different latent parts $\{P_{t_i}\}_{i=1}^m$ with a projector. Then the latent frame $Q_\tau$ is interpolated with the affine transformations estimated by the AffNet. Sequentially, a bi-branched decoder renders $R_\tau$ from $Q_\tau$.

blur kernel with the appearance and shape of the object. However, a simple convolution kernel cannot capture the delicate variance of a moving object. To this end, TbD-3D Rozumnyi et al. (2020) has approximated the integral with a piecewise linear model and leveraged energy minimization to solve the deblatting task. Specifically, the blurring stripe is considered as the sum of several small blurs according to a partition of the exposure time and each small blur is approximated by a linear motion blur. Since the linear approximation is performed in the image space directly, it cannot well capture motions with rotation when the shapes of the FMOs are complex. Furthermore, the inference time consumption caused by the energy minimization method is usually prohibitively expensive.

On the other hand, DeFMO Rozumnyi et al. (2021a) has firstly proposed to solve the deblatting task with a deep generative structure, which embeds the blurred input into a latent space representation. In particular, it adopts shared latent embedding for different time index and directly concatenates the specific time index with the embedding as intermediate features for decoding, as shown in Figure 1 (a). With a large-scale training dataset, DeFMO has achieved better performance than previous methods. However, since large relative motions inevitably exist between images from different time indexes, such shared embedding setup might not fully capture the time-varying property of a moving object. This would limit the flexibility of generating high-quality sequences with motion trajectory consistency.

To address the limitations mentioned above, we propose a novel Latent Decomposition-Interpolation Network (LDINet) to elaborately construct varying latent representations for different time indexes, which encourages to capture the inherent property of a moving object and helps generate the satisfying image sequence. In particular, a simple yet effective Decomposition-Interpolation Module (DIM) is introduced in the neural network model for FMOs deblatting, where the feature maps in the latent space output by the encoder are decomposed into several latent parts according to a fine partition of the exposure time, as shown in Figure 1 (b). For each time index in the exposure time interval, an interpolation method with affine transformations is proposed to aggregate the adjacent parts into a latent frame, which is decoded with a bi-branched decoder to predict the corresponding appearance and mask on the time index. Here, the affine transformations are well-desgined to enforce motion continuity and consistency for the generated sequence. Specifically, considering that the convolutional encoder introduces different coordinate-dependent components in the feature maps and the processing ways of these components are different under affine transformation, we propose to split the latent feature map into a scalar part and a gradient part for the decomposition-interpolation process, which helps release the full potential of affine transformation.

In addition, a Refining Conditional Deblatting (RCD) approach is presented by using post hoc image-to-image method to further enhance the fidelity of the textures and the accuracy of the masks. Specifically, this model uses the output preliminarily generated by LDINet as a condition to refine the quality of the synthesized sequence. In particular, multi-scale feature maps extracted from the preliminary results are fused with those of the original background and the blurred fast-moving object pictures to guide the model in mask repair and consistency enhancement.

Extensive comparison experiments and ablation studies are conducted to show the effectiveness of our designs for FMOs deblatting and our LDINet and RCD have achieved competitive performance compared to the existing competing methods.

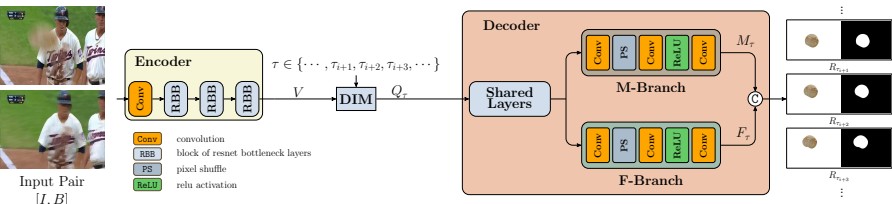

Figure 2: **A detailed illustration of LDINet.** For LDINet, the input pair $I, B$ is first processed by the encoder to obtain the feature maps $V$. Then DIM synthesizes the latent frame $Q_\tau$ at time index $\tau$ from $V$. Finally, based on $Q_\tau$, the bi-branched decoder generates the rendering $R_\tau$ consisting of the mask $M_\tau$ and the appearance $F_\tau$.

## 1 DEBLATTING METHOD

In this section, we first introduce the task setting of FMOs deblatting and give an overview of the proposed LDINet. Then the well designed Decomposition-Interpolation Module (DIM) in our LDINet is elaborated in Section 1.1, and the detailed structures of LDINet are provided in Section 1.2. Further, the corresponding learning objectives are presented in Section 1.3. Additionally, a Refining Conditional Deblatting (RCD) approach based on the post-image-to-image technique is introduced in Section 1.4.

**Preliminary**. Given the appearance $F_t$ and the mask $M_t$ of the moving object at any time $t$ within the exposure time which is rescaled to $[0, 1]$, the resultant blurred FMOs image $I$ can be formulated as

$$I = \int_0^1 F_t \, M_t + (1 - M_t)B \, \mathrm{d}t, \tag{1}$$

where $B$ is the background. However, in the deblatting task for FMOs, based on a blurred image $I$ and an estimated background $B$, the goal is to approximate a sharp rendering $R_\tau = [F_\tau \, M_\tau]$ at any given time index $\tau \in [0, 1]$. Here both the image $I : D \to \mathbb{R}^3$ and background $B : D \to \mathbb{R}^3$ are RGB images where $D \subset \mathbb{R}^2$ is the canvas consisting of $H \times W$ pixels. The rendering $R_\tau : D \to \mathbb{R}^4$ is an RGBA image where the RGB part is the appearance $F_\tau$ and the alpha part is the mask $M_\tau$. Our estimation for the rendering $R_\tau$ is denoted as $\hat{R}_\tau = [\hat{F}_\tau \, \hat{M}_\tau]$. Besides, during training, the renderings $\{R_{\tau_i}\}_{i=1}^n$ of equally spaced time indexes $\{\tau_i\}_{i=1}^n$ are available in the dataset, where $\tau_i = \frac{i-1}{n-1}$.

**The overview of LDINet**. As shown in Figure 2, the proposed LDINet is composed of an encoder, a DIM, and a decoder. In particular, the encoder first takes a blurred image $I$ and a background image $B$ as input and outputs feature maps $V$. Then a Decomposition-Interpolation Module (DIM) is introduced to decompose $V$ and to interpolate the target latent frame $Q_\tau$ for the given time index $\tau$, which would be further explained in the following section. Finally, the decoder generates the rendering $R_\tau$ with the target latent frame $Q_\tau$. The decoder is composed of several shared layers and two branches, which estimate the mask $M_\tau$ and appearance $F_\tau$ separately.

### 1.1 THE DECOMPOSITION-INTERPOLATION MODULE

Compared with the conventional deblurring tasks, the main differences of the FMOs deblurring task are the longer blurred stripe and more complex motion trajectory of the object, which makes it difficult to be resolved. However, if we consider a small time segment $\Delta t$ of the total exposure time interval $\Delta T$, the size of the blurred stripe within this time segment is small and the motion of the object is much simpler, which can be approximated by a linear motion as in Rozumnyi et al. (2020). From this point of view, the blur formation model in Equation 1 can be reformulated as

$$I = \sum_{k=0}^{m-1} \int_{k\Delta t}^{(k+1)\Delta t} F_\tau M_\tau + (1 - M_\tau)B\mathrm{d}\tau \approx \frac{1}{m} \sum_{k=0}^{m-1} H_{t_k} \otimes F_{t_k} + (1 - H_{t_k} \otimes M_{t_k})B, \tag{2}$$

where $\Delta t = \frac{1}{m}$, $\otimes$ is the convolution operation, and $H_{t_k}$ is the kernel containing the motion information around the time index $t_k = \frac{k-1}{m-1}$.

Inspired by this observation, we consider that the feature maps could also be decomposed into a set of parts corresponding to a series of discrete time indexes. Then the latent frame of the target time index

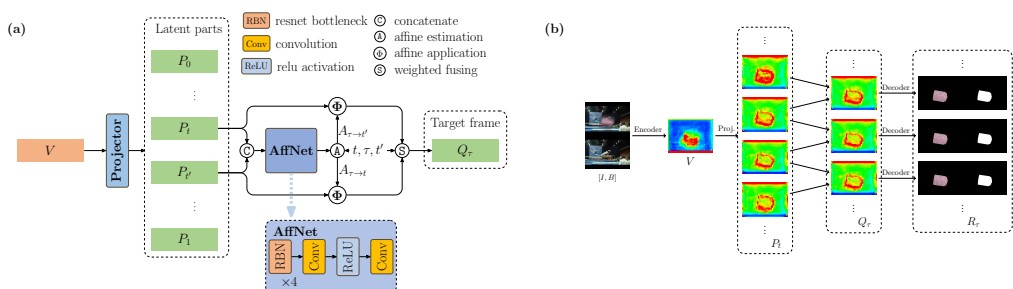

Figure 3: **The decomposition-interpolation module (DIM)**. (a) An overview of the decomposition-interpolation module. (b) Visualization of the intermediate results of DIM with the silhouettes in heatmaps.

can be obtained by interpolation. Thus, a Decomposition-Interpolation Module (DIM) is proposed to explore the structure of the latent space more appropriately to generate a better latent frame for the target time index, as shown in Figure 3 (a). In particular, the feature maps $V$ are first decomposed into $m$ latent parts $\{P_{t_i}\}_{i=1}^m$ in the latent space corresponding to the $m$ discrete time indexes $\{t_i\}_{i=1}^m$ with a projector. Here we assume that the part $P_{t_i}$ contains the motion and appearance information of the object around the time index $t_i$. Then given the target time index $\tau$, the required latent frame $Q_\tau$ can be synthesized with $\{P_{t_i}\}_{i=1}^m$ by interpolation.

Further, since the FMOs we deal with are mostly rigid objects, the changes of the appearances of moving object that are adjacent in time indexes can be modeled by simple affine transformations in the image space. However, while in the deep latent space, the complex mapping of the encoder would introduce nonlinear behaviors for the affine transformations. In our case where the encoder is a convolution network, for a single input channel, the convolution operation could be regarded as a linear combination of summation and series of directional derivatives. Since the directional derivatives are linear projections of the gradient fields, it is reasonable to represent the convolution results by scalar fields and gradient fields. More explanation is provided in Appendix A.

Moreover, as the processing ways of scalar and gradient fields under the affine coordinate transformations is different, the latent part $P_t$ is divided into scalar fields $P_t'$ and gradient fields $P_t''$. In particular, for the scalar fields, the transformation behavior is the same as the affine transformation in the image space,

$$P_\tau'(\mathbf{x}) = P_t'(A(\mathbf{x})), \tag{3}$$

where $A(\cdot)$ is the affine transformation describing the motion from time index $\tau$ to $t$ on the point $\mathbf{x}$ in 2D coordinate. While for the gradient fields, the behavior of the transformation becomes

$$P_\tau''(\mathbf{x}) = P_t''(A(\mathbf{x}))\tilde{A}, \tag{4}$$

where $\tilde{A}$ is the Jacobian of $A(\cdot)$. Specifically, we first suppose that each latent part $P_t$ could be represented by a concatenation of the scalar fields $P_t'$ and the gradient fields $P_t''$, i.e., $P_t = [P_t'\ P_t'']$, and thus they would be processed separately. Then we denote $\mathbf{\Phi}[A, P_t]$ as an operator which applies the affine transformation $A$ to the latent part $P_t$ in a way that the scalar fields $P_t'$ and the gradient fields $P_t''$ are first transformed by Equation 3 and Equation 4 respectively, and then concatenated as the result. In this way, the latent frame of the target time index would be approximated more appropriately.

Based on the above transformation method, one remaining difficulty is how to estimate the affine transformation in the feature space. To this end, we introduce a residual network named AffNet, as shown in Figure 3 (a), which takes two latent parts as input and predicts a pair of affine transformations between them. Since there are several downsampling layers in the encoder, the feature maps shrink several times in size. Thus each grid in the feature maps indeed contains the information of a patch of the input image. Therefore, we predict the affine transformations point-wisely that the AffNet generates affine transformations for each grid of the latent parts separately.

Finally, given the affine transformations estimated by AffNet, our interpolation process is presented more formally. In particular, as shown in Figure 3 (a), to interpolate the latent frame $Q_\tau$ at the time index $\tau \in [0, 1]$, we first find the two nearest latent parts $P_t$ and $P_{t'}$ from the decomposition results, which satisfy $t \le \tau \le t'$, to obtain the affine transformations $A_{t \to t'}$ and $A_{t' \to t}$ between these two parts by AffNet. Next, to obtain the affine transformations from time $\tau$ to $t$ and from time $\tau$ to $t'$, we approximate them by $A_{\tau \to t} = I + \frac{\tau - t}{t' - t}(A_{t' \to t} - I)$ and $A_{\tau \to t'} = I + \frac{t' - \tau}{t' - t}(A_{t \to t'} - I)$, respectively. Then the target latent frame is interpolated with the affine transformations as

$$Q_\tau = \frac{t' - \tau}{t' - t} \mathbf{\Phi}[A_{\tau \to t}, P_t] + \frac{\tau - t}{t' - t} \mathbf{\Phi}[A_{\tau \to t'}, P_{t'}]. \tag{5}$$

Note that a weighting scheme is employed here to fuse the information from both of the two neighboring parts. Besides, the application of $\mathbf{\Phi}$ is along the grids of the latent parts with the corresponding affine transformations point-wisely.

To provide an intuitive picture of DIM, we visualize the intermediate results of DIM in Figure 3 (b). As we can see in the figure, the input pair is first encoded into an embedding in the feature space by the encoder. In this process, the information of background is discarded while the blurry part is remained. Then DIM decomposed the embedding to several latent parts where each latent part contains the information of its piece of time interval which can be verified by the silhouettes in heatmaps. And the interpolation operation of DIM generates the latent frames of the specific time indexes. Finally, the decoder generates the target masks and appearances with the latent frames.

## 1.2 THE STRUCTURES OF THE LDINET

**The structure of the Encoder**. The structure of the encoder is based on the ResNet-50 He et al. (2016) with the nuance that we only take the first three downsampling blocks and extend the last block with five ResNet bottlenecks. The channel number of the feature map generated by the encoder is 1024.

**The structure of the Decoder**. As shown in Figure 2, the decoder is composed of several shared layers and two convolutional branches. To be specific, the shared layers are two residual blocks. Each residual block is followed by a pixel shuffle layer Shi et al. (2016) which up-scales the spatial size of the latent frame by a factor of two. The output channel numbers of the residual blocks are 256 and 64, respectively. Given the up-scaled latent frame, we use two convolutional branches to estimate the RGB channels for appearance and the alpha channel for mask respectively. These two branches are similar in structure. In each branch, we first use a $3 \times 3$-convolution layer with 64 output channels. Then a pixel shuffle layer is applied to up-scale the size of the feature maps by a factor of two. Finally, the feature maps go through two convolutional layers with the numbers of output channels being 16 and 4 respectively and are transformed into outputs with the last layer.

**The structure of DIM**. Here we introduce the network structure in DIM, including the projector and the AffNet. The projector is a ResNet bottleneck block and its number of output channels is $512m$ where $m$ is the number of the output parts. The AffNet accepts an input with 1024 channels which is concatenated by two latent parts. The structure of AffNet is shown in Figure 3 (a). The first ResNet bottleneck block reduces the channel number to 64. And the rest three ResNet bottleneck blocks keep the channel number unchanged. Finally, the predictor first reduces the channel number from 64 to 16 with the first $3 \times 3$-convolution layer. After a ReLU activation layer, the second $3 \times 3$-convolution layer predicts 6 parameters for each affine transformation.

## 1.3 THE TRAINING LOSS

In this section, we introduce the training objectives of our LDINet, which can be divided into two categories according to the space where the constraints are performed, i.e., the image and the latent space. In particular, in the image space, a reconstruction loss $\mathcal{L}_R$ is introduced to reconstruct the masks, the appearances, and the blurry input. Besides, a sharpness loss $\mathcal{L}_S$ is employed to sharpen the masks. As for the latent space, three objectives $\mathcal{L}_L$, $\mathcal{L}_{id}$, and $\mathcal{L}_C$ are introduced to encourage the feature invariance to different backgrounds, stabilize the training of the AffNet, and improve the feature consistency between adjacent latent parts, respectively.

**Direction of motion trajectory**. Before introducing the details of the loss functions, we first clarify the correspondence between the predicted sequence $\{\hat{R}_{\tau_i}\}_{i=1}^n$ and the ground truth $\{R_{\tau_i}\}_{i=1}^n$.

Specifically, since the motion blur keeps invariant when the motion trajectory is reversed, the direction of the motion trajectory is ambiguous in fact. In order to determine the direction of the motion trajectory, we use the relative error rate of the masks

$$\text{Err}(\hat{R}, R) = \sum_{\tau} \frac{\sum_{p \in D} |\hat{M}_\tau^p - M_\tau^p|}{\sum_{p \in D} M_\tau^p}, \tag{6}$$

as the criteria and select the direction with a smaller relative error rate, where $p \in D$ runs over the pixels of the canvas $D$. For simplicity, with some abuse of notation, $\{\hat{R}_{\tau_i}\}_{i=1}^n$ is used in the following description to represent the estimated rendering sequence in the selected direction.

**Reconstruction loss**. The reconstruction of the rendering at a given time index consists of three parts, i.e., the reconstruction of the mask, the appearance, and the blurry input. We use Binary Cross Entropy (BCE) loss for the reconstruction of the mask and L1 loss for the reconstruction of the appearance and the input. In particular, for the reconstruction of the appearance, the constraint is performed between the estimated and ground truth instance images $\hat{I}_\tau = \hat{M}_\tau \hat{F}_\tau + (1 - \hat{M}_\tau)B$ and $I_\tau = M_\tau F_\tau + (1 - M_\tau)B$ instead of between the estimated and ground truth appearances $\hat{F}_\tau$ and $F_\tau$. As for the reconstruction of the blurry input, it encourages the consistency between the rendering model and the formation model of the blurry input in a self-supervised manner, where the estimation of the blurry input $\hat{I} = \frac{1}{n} \sum_\tau \hat{I}_\tau$ is enforced to match the blur input image. Besides, a shape-aware weighting scheme $W_\tau$ is further presented to reweight the appearance loss of each pixel based on its location to the outline of the object. In practice, the weighting scheme is obtained by blurring the mask $M_\tau$ with an average kernel $K_{avg}$, i.e., $W_\tau = K_{avg} \otimes M_\tau$. Thus the overall reconstruction loss $\mathcal{L}_R$ is

$$\mathcal{L}_R = \frac{1}{n} \sum_{\tau} \frac{\sum_{p \in D} \left( \ell_{\text{BCE}}(\hat{M}_\tau^p, M_\tau^p) + W_\tau^p \ell_1(\hat{I}_\tau^p, I_\tau^p) \right)}{\sum_{p \in D} M_\tau^p} + \frac{\sum_{p \in D} \ell_1(\hat{I}^p, I^p)}{\sum_{p \in D} \mathbb{1}[(\sum_\tau M_\tau^p) > 0]}, \tag{7}$$

where $\ell_{\text{BCE}}$ is the point-wise BCE loss, $\ell_1$ is the point-wise L1 loss, and $\mathbb{1}[\cdot > 0]$ is an indicator function which assigns 1 to the positive values and 0 to the others.

**Mask sharpening loss**. To sharpen the predicted masks, we strengthen the correct prediction results in the estimated masks by decreasing the prediction entropy for the correctly classified pixels,

$$\mathcal{L}_S = \frac{1}{n|D|} \sum_{\tau} \sum_{p \in D} \mathbb{H}(\hat{M}_\tau^p G_\tau^p), \tag{8}$$

where $G_\tau$ is a binary map that indicates the correctly classified pixels and $\mathbb{H}$ is the point-wise binary entropy.

**Background reduction loss**. Considering that the rendering results should be invariant to the change of background, $\mathcal{L}_L$ is designed to reduce the influence of background on the feature maps. Specifically, given two inputs which only differ in the background, they are first encoded as $V$ and $V'$ in the latent space by the encoder and then constrained by

$$\mathcal{L}_L = \|V - V'\|_2^2. \tag{9}$$

**Feature consistency between latent parts**. Since the motion trajectories of the objects are continuous, we consider that the latent frames should also be similar when the corresponding time indexes are close. Since the latent frames are interpolated from the latent parts, we formulate the feature consistency loss $\mathcal{L}_C$ as

$$\mathcal{L}_C = \frac{1}{m-1} \sum_{i=1}^{m-1} \|P_{t_i} - P_{t_{i+1}}\|_2^2. \tag{10}$$

**Reversibility of the affine transformations**. Since the output of the AffNet is a pair of forward and backward affine transformations between the two input latent parts, we intend to constrain the two affine transformations to be the inverse of each other,

$$\mathcal{L}_{id} = \|A_{t \to t'} A_{t' \to t} - I\|_F^2, \tag{11}$$

where $\|\cdot\|_F$ is the Frobenius norm, $I$ is the identity matrix and $A_{t\to t'}$ and $A_{t'\to t}$ are the forward and backward transformations between the two latent parts at the time indexes $t$ and $t'$ estimated by AffNet, respectively.

**Joint loss**. Consequently, the joint loss function is a combination of the two aspects,

$$\mathcal{L}_{joint} = \underbrace{\mathcal{L}_R + \mathcal{L}_S}_{\text{image space}} + \underbrace{\mathcal{L}_{id} + \mathcal{L}_L + \alpha_C \mathcal{L}_C}_{\text{latent space}}. \tag{12}$$

### 1.4 Refining Conditional Deblatting

Following the LDINet, we obtain a sequence of preliminary outputs which contains the structures of motion and appearance of the object. Inspired by the success of the "coarse-to-fine" schemes in existing image deblurring Tao et al. (2018), a Refining Conditional Deblatting (RCD) approach is introduced to further enhance the the fidelity of the textures and the accuracy of the masks by leveraging the initially generated results, as shown in Figure 4. In particular, we select the first frame $R_0^c$, the middle frame $R_{0.5}^c$ and the last frame $R_1^c$ from the generated output and concatenate them in the channel dimension as the condition of this refining model. The architecture of RCD is also based on LDINet and only the encoder is modified to

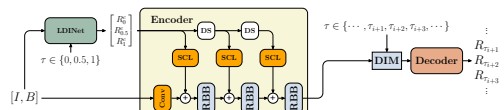

Figure 4: **The Pipeline of the RCD**. For this refining method, we concatenate the first, the median and the last frames generated from the LDINet as the conditional inputs. And we use two DownSampling (DS) layers to obtain the conditional information in different scales and embed them with Simple Convolution Layers (SCL) to make fusion before each blocks of the encoder.

efficiently fuse the condition frames in a multi-scale fashion. To be specific, we first downscale the frames two times by a factor of two to obtain two smaller scales of condition frames. Here pixel unshuffle layers are chosen as the downscaling method to keep the most information of the conditions. Then each scale of condition is separately embedded to feature representation through a simple convolution layer which is sequentially stacked by a convolution layer, a batchnorm layer and a ReLU layer. Finally, the multi-scale features are added to the inputs before each block of the encoder according to their scales so that the condition frames are fused into the model.

## 2 Experiments

In this section, the training and evaluation datasets are first introduced in Section 2.1 and 2.2 and the training details are provided in Section 2.3. Then the proposed LDINet and its refining version RCD are compared with the existing state-of-the-art methods in Section 2.4. Further, extensive ablation studies are conducted to evaluate the effect of each component in LDINet in Section 2.5.

### 2.1 Synthesized training dataset

The synthesized dataset for training is based on the training dataset of DeFMO Rozumnyi et al. (2021a), which is generated with Blender Cycles Community (2018). Each training sample is created by a 3D object moving through a 6D linear trajectory over two background sequences and consists of two backgrounds for background reduction, one FMOs blur stripe for the construction of blurry inputs, and 25 discrete frames of sharp renderings of the object at equally spaced time indexes within the exposure time $[0, 1]$ including the start and end time. Following Kotera et al. (2019), static backgrounds are employed here. The 3D objects are sampled from ShapeNet Chang et al. (2015) dataset applied with DTD Cimpoi et al. (2014) textures. The backgrounds for training are sampled from the VOT Kristan et al. (2016) sequences, and the backgrounds for validation are sampled from Sports-1M Karpathy et al. (2014). There are 50,000 samples for training and 1,000 samples for validation.

## 2.2 EVALUATION DATASET

The evaluation datasets are three real-world datasets from the FMOs deblatting benchmark Rozumnyi et al. (2021a):

**TbD Kotera et al. (2019)** is composed of 12 sports sequences with uniformly colored and mostly spherical objects. Each sequence contains 16 ~ 60 frames.

**TbD-3D Rozumnyi et al. (2020)** is composed of 10 sequences and contains objects with complex textures, which makes it more difficult. Each sequence contains 37 ~ 81 frames. The rotations of the objects result in significant differences in their appearances. One limitation is that the objects are mostly spherical, so their shapes remain constant when rotated.

**Falling Objects Kotera et al. (2020)** is composed of 6 sequences and is the most challenging benchmark with objects of complex textures and 3D shapes. Each sequence contains 11 ~ 22 frames.

For each dataset, the low-speed sequences are created by averaging over the full exposure high-speed ground truths. The ground truths have a frame rate that is 8 times higher than that of the low-speed sequences.

## 2.3 TRAINING SETTINGS

The training of LDINet contains two stages, a warm-up stage and a finetuning stage. In the warm-up stage, we aim to train the AffNet and provide a guidance to disentangle the original latent part into the scalar part and the gradient part. In particular, since DIM is not well trained at beginning, we use the interpolation method with weighted summation over the latent parts during the warm-up. To train the AffNet, we first introduce a pseudo input by applying a small random affine transformation to the FMOs blur stripe in the image space. Then the latent parts of the original input and the pseudo input are fed into the AffNet to estimate this random affine transformation. Furthermore, a consistency is introduced to constraint between the transformation in the image space and the transformation in the latent space. Please refer to Appendix B for more details. In the finetuning stage, we train the model for 20 epochs in total. For the first 10 epochs, we use the learning rate $lr = 1e - 4$ and set $\alpha_C = 0.01$. The learning rate is reduced to $1e - 5$ and $\alpha_C = 0$ is set for the rest 10 epochs. As for the refining version RCD method, we use the parameters of the LDINet to initialize the conditional model and train the model for 25 epochs. The learning rate is reduced from $1e - 4$ to $1e - 6$ with cosine annealing Loshchilov & Hutter (2016). During the training processes, the part number of DIM is $m = 16$, and the kernel size of the average kernel $K_{avg}$ is $11 \times 11$. We use Adam optimizer Kingma & Ba (2015) with batch size 24. The model is trained on 8 Nvidia A5000 GPUs and the total training time is about 1.5 days for LDINet. The average results of three runs are reported.

## 2.4 EVALUATION

In this section, we compare the proposed methods with the state-of-the-art methods on a variety of datasets. To be specific, we first compare them with the existing FMOs deblatting methods based on energy minimization (TbD Kotera et al. (2019) and TbD-3D Rozumnyi et al. (2020)) and the data-driven methods (DeFMO Rozumnyi et al. (2021a) and BiT++Zhong et al. (2023)). Note that BiT++ only predicts the sharp images, and thus we do not report its trajectory estimation results. Besides, we provide the results of our model with SfB Rozumnyi et al. (2021b) in Appendix E. The Peak Signal-to-Noise Ratio (PSNR), Structure Similarity Index Measure (SSIM), and Trajectory Intersection over Union (TIoU) are chosen as the evaluation metrics. Following the protocols from DeFMO Rozumnyi et al. (2021a), we generate the estimation of the ground truths by averaging over the sequences every 5 frames to match the exposure time of the ground truths in the evaluation datasets. Considering the ambiguity of the direction of motion trajectory, we choose the direction with a better PSNR score. The sub-frame trajectory is estimated using the center of the generated estimation of mask $\hat{M}_\tau$.

The evaluation results are provided in Table 1. It can be observed that the data-driven methods outperform the energy-minimization methods by a wide margin and the performance gap increases as the shapes of the objects in the datasets become more complex (e.g., the Falling dataset). We speculate that this is primarily due to the limitations of the prior assumptions used in the energy-minimization methods. As the objects' shapes become more complex, these prior assumptions no longer match the

Table 1: **Comparison results of different methods on the FMOs deblatting task**. The best results are marked in bold and the second best results are underlined. For our proposed methods, we run the model 3 times and report the standard deviation results in the parentheses. The running time analysis is provided in Appendix D.

| Dataset | Score | Compared Methods | | | | The proposed | |
|---|---|---|---|---|---|---|---|
| | | TbD | TbD-3D | BiT++ | DeFMO | LDINet | RCD |
| Falling | TIoU↑ | 0.539 | 0.539 | N/A | 0.684 | **0.686**(.007) | 0.684(.006) |
| | PSNR↑ | 20.53 | 23.42 | 25.62 | 26.83 | 28.09(0.01) | **28.36**(0.01) |
| | SSIM↑ | 0.591 | 0.671 | 0.704 | 0.753 | 0.771(.001) | **0.779**(.002) |
| TbD-3D | TIoU↑ | 0.598 | 0.598 | N/A | 0.879 | 0.906(.002) | **0.908**(.001) |
| | PSNR↑ | 18.84 | 23.13 | 25.86 | 26.23 | 26.50(0.14) | **26.64**(0.06) |
| | SSIM↑ | 0.504 | 0.651 | 0.662 | 0.699 | 0.707(.003) | **0.713**(.002) |
| TbD | TIoU↑ | 0.541 | 0.542 | N/A | 0.550 | 0.616(.004) | **0.630**(.009) |
| | PSNR↑ | 23.22 | 25.21 | 24.93 | **25.57** | 25.24(0.12) | 25.55(0.13) |
| | SSIM↑ | 0.605 | **0.674** | 0.573 | 0.602 | 0.626(.008) | 0.631(.002) |

|  I/B  |  DeFMO  |  LDINet  |  RCD  |  GT  |

Figure 5: **Deblatting results**. The leftmost column shows the input pairs, the blurred image $I$, and the background $B$. The rightmost column shows the ground truth. We represent the results for the shape key from the dataset Falling Objects Kotera et al. (2020). For each method, we show the estimated appearance (left), the estimated mask (right) and the temporal super-resolution frames at $t = 0$ (top) and $t = 1$ (down).

distribution of the datasets, resulting in bias errors. This also suggests that the data-driven methods could derive a more precise prior from the training data. Then compared to the existing methods, our LDINet achieve better performances in most cases on all the three datasets by introducing the decomposition-interpolation module in the latent space. In particular, on the Falling Objects Kotera et al. (2020), LDINet outperforms DeFMO by 1.26 dB on the PSNR metric, demonstrating that the proposed method can better solve the fast-moving objects with complex shapes. On the TbD-3D and TbD datasets, LDINet also outperforms DeFMO in most cases. Further, the results of RCD outperform those of LDINet, which verifies the effectiveness of the refining framework with preliminarily generated results as condition. In addition, besides the above comparison under the setting of static backgrounds as Kotera et al. (2019); Rozumnyi et al. (2021a), we also provide analysis under background shifts. Please refer to Appendix F for more details.

In addition, some qualitative deblatting results are given in Figure 5. It is shown that the masks of the falling key generated by LDINet and RCD have higher qualities than DeFMO, and the appearances are also more precise in our results compared to DeFMO. Please refer to Appendix C for more visualization results.

## 2.5 ABLATION STUDY

In this section, we conduct ablation studies to analyze the effects of different components and the hyperparameters in the proposed LDINet.

As shown in Table 2, we can see that warm-up stage is necessary to improve the effectiveness of AffNet and the performance drops by replacing the affine transformation with the linear interpolation. Besides, the introduction of the bi-branched structure provides significant improvements on the metrics, by separating the estimation of the appearance and the mask. On the other hand, it is seen that reducing the influence of the background on the feature maps with $L_L$ shows a positive impact. However, lacking the regularization term $\mathcal{L}_C$ between the adjacent latent parts that are decomposed in DIM results in a significant drop in the performance of the model. Moreover, without the reversible term $\mathcal{L}_{id}$ that keeps the affine transformation in two directions to be the inverse of each other, the metrics show a slight drop on the Falling Objects dataset but a relatively large drop on the TbD dataset. This indicates that this term would provide some regularization for the prediction of the affine transformations and reduce overfitting on the training set. Finally, the weighting scheme $W_\tau$ also

Table 2: **Ablation study: warm-up, architecture, and objectives**. The effects of the warm up, the interpolation method, the structure of the decoder, the introduction of the reversible loss $\mathcal{L}_{id}$, the background reduction loss $\mathcal{L}_L$, the frame consistency loss $\mathcal{L}_C$, and the weighting scheme $W_\tau$ are investigated on Falling Objects and TbD datasets. For the interpolation method, 'A' denotes using affine transformation in the interpolation, 'L' denotes using linear interpolation, and 'T' denotes concatenating time indexes as DeFMO. The best results are marked in bold.

| Warmup | Arch. | | Objective | | | | Falling Objects | | | TbD | | |
|---|---|---|---|---|---|---|---|---|---|---|---|---|
| | interp. | bi-branched | $\mathcal{L}_{id}$ | $\mathcal{L}_L$ | $\mathcal{L}_C$ | $W_\tau$ | TIoU↑ | PSNR↑ | SSIM↑ | TIoU↑ | PSNR↑ | SSIM↑ |
| - | T | ✓ | - | ✓ | - | ✓ | 0.679 | 27.06 | 0.741 | 0.554 | 24.97 | 0.609 |
| - | L | ✓ | - | ✓ | ✓ | ✓ | 0.681 | 27.64 | 0.762 | 0.603 | 25.20 | 0.615 |
| ✗ | A | ✓ | ✗ | ✓ | ✓ | ✓ | 0.655 | 27.73 | 0.761 | 0.571 | 25.58 | 0.593 |
| ✗ | A | ✓ | ✓ | ✓ | ✓ | ✓ | **0.697** | 27.71 | 0.766 | 0.615 | **25.70** | 0.626 |
| ✓ | A | ✗ | ✓ | ✓ | ✓ | ✓ | 0.689 | 27.39 | 0.755 | 0.590 | 24.74 | 0.598 |
| ✓ | A | ✓ | ✓ | ✗ | ✓ | ✓ | 0.687 | 27.66 | 0.767 | 0.606 | 23.79 | 0.577 |
| ✓ | A | ✓ | ✓ | ✓ | ✗ | ✓ | 0.681 | 27.50 | 0.763 | 0.605 | 23.80 | 0.581 |
| ✓ | A | ✓ | ✓ | ✓ | ✓ | ✗ | 0.679 | 27.34 | 0.753 | 0.612 | 24.44 | 0.592 |
| ✓ | A | ✓ | ✓ | ✓ | ✓ | ✓ | 0.686 | **28.09** | **0.771** | **0.616** | 25.24 | **0.626** |

improves the performance of the model by decreasing the supervision strength for the error-prone area (i.e., the border of the objects) and pays more attention to the inner pixels of the object that are precisely segmented by the estimated mask.

Next, we investigate the effects of the number of latent parts in the decomposition and the proportion of the scalar channels for the interpolation of DIM. The results are provided in Table 3 and Table 4. First, the number of parts in the decompostion of DIM controls the fidelity of DIM. In particular, with more parts, the time interval between adjacent parts becomes smaller and the transformation between the adjacent parts behaves more likely to a linear transformation, which improves the affine estimation quality. As shown in Table 3, increasing the number of parts in DIM structure brings a better performance. However, we note that the increase of the number of parts also leads to the explosion of the memory footprint and a heavy calculation burden. Thus, we set the number of parts to 16 in our implementation of DIM. On the other hand, the effect of disentangling the latent part into the scalar fields and the gradient fields is shown in Table 4. It is seen that the totally scalar-like (i.e., the proportion is 1) or totally gradient-like (i.e., the proportion is 0) latent part does not obtain the best performance. This indicates that simply formulating the latent parts as scalar fields or gradient fields is not enough to capture the complex transformation behavior in the latent space under the affine transformation, while the mixture of the scalar fields and gradient fields provides a more appropriate approximation.

Table 3: **Ablation study: the number of latent parts.** The evaluation is conducted on the Falling Objects dataset. The best results are marked in bold.

| Number of parts | Falling Objects | | |
|---|---|---|---|
| | TIoU↑ | PSNR↑ | SSIM↑ |
| 4 | 0.683 | 27.49 | 0.748 |
| 8 | 0.689 | 27.82 | 0.757 |
| 12 | **0.694** | 27.90 | 0.765 |
| 16 | 0.686 | **28.09** | **0.771** |
| 20 | 0.688 | 27.89 | 0.767 |

Table 4: **Ablation study: the proportion of scalar channels.** The evaluation is conducted on the Falling Objects dataset. The best results are marked in bold.

| Proportion | Falling Objects | | |
|---|---|---|---|
| | TIoU↑ | PSNR↑ | SSIM↑ |
| 1 | 0.684 | 27.76 | 0.757 |
| 1/2 | 0.669 | 27.90 | 0.769 |
| 1/3 | **0.686** | **28.09** | **0.771** |
| 0 | 0.686 | 27.78 | 0.762 |

# 3 CONCLUSION

In this paper, we propose a new LDINet for single image FMOs deblatting. In particular, we introduce a decomposition-interpolation module in the latent space which first decomposes the feature maps into several latent parts to incorporate the prior of the temporal sequential structure into the deblatting process and then aggregates the adjacent parts with affine transformations to properly interpolate the target latent frames for each time index. Further, we present a refining conditional deblatting method based on the results of LDINet to further enhance the output quality. Extensive experiments are conducted and the evaluation results show that our LDINet and RCD have achieved superior performances in most cases when compared with the existing methods.

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

## A  THE EXPLANATION ON THE INTRODUCTION OF SCALAR FIELDS AND GRADIENT FIELDS

Taking a simple convolution operation Conv with $3 \times 3$ kernel size for example, the convolution result over a single channel input can be regarded as a linear combination of a scale operation and eight directional derivatives,

$$
\begin{aligned}
\text{Conv}[I](x,y) &= \sum_{i,j\in\{-1,0,1\}} k_{i,j} I(x+i, y+j) \\
&= \sum_{i,j\in\{-1,0,1\}} k_{i,j}((I(x+i,y+j) - I(x,y)) + I(x,y)) \\
&= \sum_{i,j\in\{-1,0,1\}} k_{i,j}(D_{i,j}[I](x,y) + I(x,y)),
\end{aligned}
\tag{13}
$$

where $k_{i,j}$ are the kernel coefficients, $D_{i,j}$ are the image directional derivatives. With a further approximation, we have

$$\text{Conv}[I](\mathbf{x}) \approx kI(\mathbf{x}) + \frac{\partial I(\mathbf{x})}{\partial \mathbf{x}} \mathbf{h}, \tag{14}$$

where $\text{Conv}[\cdot]$ is the convolution operation, $\mathbf{x}$ is the coordinate vector, $k$ is the scale, and $\mathbf{h}$ is the weighted vector for the derivative components. With an affine transformation $A(\cdot)$, the original coordinate vector $\mathbf{x}$ is transformed into new coordinate vector $\mathbf{x}' = A(\mathbf{x})$. Under this new coordinate system, the convolution result of the transformed input is

$$\text{Conv}[I'](\mathbf{x}') \approx kI'(\mathbf{x}') + \frac{\partial I'(\mathbf{x}')}{\partial \mathbf{x}'} \mathbf{h} = kI(\mathbf{x}) + \left( \frac{\partial I(\mathbf{x})}{\partial \mathbf{x}} \frac{\partial \mathbf{x}}{\partial \mathbf{x}'} \right) \mathbf{h}$$
$$= kI(\mathbf{x}) + \left( \frac{\partial I(\mathbf{x})}{\partial \mathbf{x}} \tilde{A}^{-1} \right) \mathbf{h}, \tag{15}$$

where $\tilde{A}$ is the Jacobian of $A(\cdot)$. Comparing Equation 14 and Equation 15, although the scalar fields of the target frame can still be obtained by affine transformations from those of the neighboring latent parts, the gradient fields of the target frame can not be obtained in the same way since they changes with the affine transformations. Therefore, it is reasonable to represent the convolution results by scalar fields and gradient fields and aggregate the features of parts in different ways according to their categories.

## B    THE DETAILS OF WARM-UP STAGE

From the description in Section1.1, the affine transformation estimated by the AffNet is deeply involved in the interpolation procedure of DIM. Directly using an AffNet with randomly initialized parameters would generate estimates of affine transformations with large errors and thus hinders the training procedure of the whole model. Besides, guidance is needed to disentangle the original latent part into scalar part and gradient part.

To address this difficulty, we propose several strategies to warm up the training procedure.

**Interpolation with weighted summation** Specifically, the channels of the latent part are dispatched into the scalar fields and the gradient fields. Since the AffNet is not yet accurate enough and the latent space is not well constructed, the interpolation method with weighted summation is used in the warm up stage, The weighting scheme $\boldsymbol{v}(\tau)$ in the interpolation is:

$$Q_\tau = \sum_i v_{t_i}(\tau) P_{t_i}, \tag{16}$$

where $v_{t_i}(\tau) = \frac{\exp(-\sigma(t_i - \tau)^2)}{\sum_{k=1}^m \exp(-\sigma(t_k - \tau)^2)}$ is the component and $t_i = \frac{i-1}{m-1}$ is the time index. The hyper parameter $\sigma$ is used for adjusting the correlation between the latent frame $Q_\tau$ and the latent parts $P_{t_i}$.

**Pseudo supervision for AffNet** Since there is no explicit supervision signal to train the AffNet, we generate a pseudo input $I^A$ by applying a small random affine transformation $A$ to the FMOs in the input $I^O$. And we use $A$ as the ground truth to supervise the training of the AffNet which takes the latent parts with the same time index from the two input images as input. Here, we denote the latent parts of the original input $I^O$ and the transformed input $I^A$ at time index $i$ as $P_i^O$ and $P_i^A$ respectively, and denote the predicted affine transformation from the original pieces to the transformed pieces as $\hat{A}_i = \text{AffNet}(P_i^O, P_i^A)$. Thus the loss for pseudo supervision is

$$\mathcal{L}_A = \frac{1}{mW_l H_l} \sum_{i=0}^{m-1} \sum_{j=0}^{W_l-1} \sum_{k=0}^{H_l-1} \|\hat{A}_i^{j,k} - A\|_F^2, \tag{17}$$

where $W_l$ and $H_l$ are the width and height of the latent feature maps, and $\hat{A}_i^{j,k}$ is the predicted affine transformation on the position $(j, k)$.

**Consistency between the latent and the image space under affine transformation** Here, we aim to find an appropriate latent space where the features are represented as scalar fields and gradient fields. According to the different behaviors shown by the scalar fields and gradient fields under the

affine transformation, we introduce a consistency constraint which forces the transformation results of the latent parts $\Phi[A, P_i^O]$ to approach the latent parts $P_i^A$ which are generated from the inputs transformed in the image space,

$$\mathcal{L}_T = \frac{1}{m} \sum_{i=0}^{m-1} \|\Phi[A, P_i^O] - P_i^A\|_2^2. \tag{18}$$

**Joint loss for warm-up** For warm-up stage, the loss function contains the objectives for both the deblatting process and the AffNet,

$$\mathcal{L}_{warmup} = \mathcal{L}_{joint} + \mathcal{L}_A + \mathcal{L}_T. \tag{19}$$

## C QUALITATIVE COMPARISONS

In Figure 6, we show the qualitative results of different methods for the object *cell* in Falling Objects dataset, *pen* in Falling Objects dataset, and *volleyball* in TbD-3D dataset. From the figure, we first observe that comparing to DeFMO, the proposed LDINet and RCD provide better appearance reconstruction results for the FMOs. Specifically, in the results of DeFMO, there exists certain artifacts around the object, which might come from the inaccurate estimation of the masks. Moreover, the appearance of the object across different frames are quite similar in DeFMO, which indicates that DeFMO may be incapable of well modeling the appearance change of the object during the motion. In contrast, with a well-designed decomposition-interpolation scheme, the proposed methods would better capture the motion of the object and also produce a more accurate estimation for the mask. On the other hand, the comparison results between two interpolation methods (i.e., affine transformation v.s. linear interpolation) show the effectiveness of affine transformation in aggregation for latent frames. In particular, the artifacts in the results of linear interpolation based method become heavier near the two ends of the exposure time period, such as the frame 0 and 7, because of the lack of information. However, the affine transformation based method would always provide high-quality results, due to a more appropriate way for modeling the latent parts and aggregating the information from adjacent latent parts to generate the target latent frames.

## D RUNNING TIME

Table 5: **Running time of deblatting methods**. The results are reported in terms of Frames Per Second (FPS).

|  | Compared Methods | | | | The proposed | |
|---|---|---|---|---|---|---|
|  | TbD | TbD-3D | BiT++ | DeFMO | LDINet | RCD |
| FPS | <1 | <1 | 2 | 39 | 32 | 30 |

To compare the running time of the deblatting methods, we follow the protocol of the test benchmarks where we generate 8 frames for each sample with a single NVidia A5000 GPU and each frame is the average of 5 sub-frames. From the results in Table 5, LDINet and RCD show the comparable running time to DeFMO. Besides, the running time of RCD only increases about 10% compared with the result of LDINet because RCD only computes three frames of rendering results from LDINet as condition which largely reduces the computation cost compared with the case where fully 40 sub-frames are generated.

## E SfB RESULTS

To verify the quality of the results of our model for further processing, we report the results using SfB Rozumnyi et al. (2021b) in Table 6. We use the 8-step and learning rate 0.03 setting in SfB. The results show that our methods give more acceptable estimation quality comparing with DeFMO.

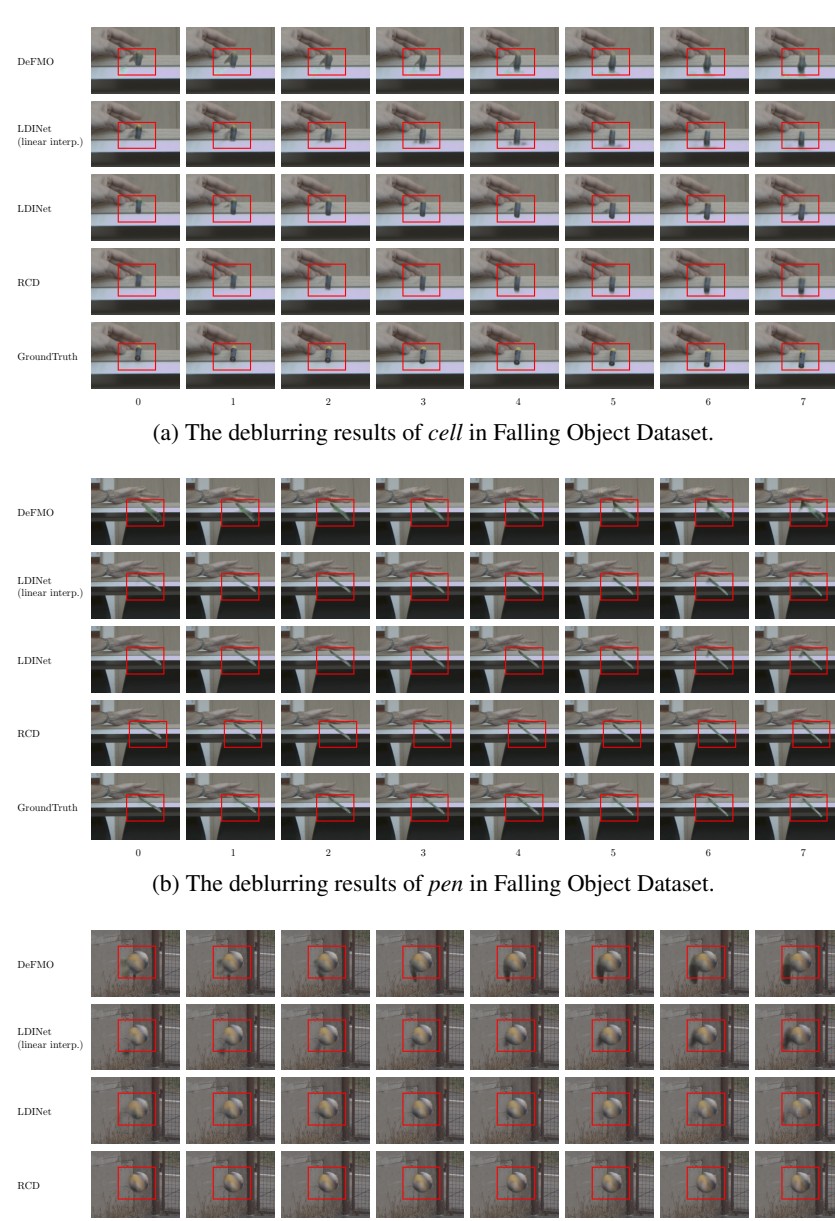

(a) The deblurring results of *cell* in Falling Object Dataset.

(b) The deblurring results of *pen* in Falling Object Dataset.

(c) The deblurring results of a moving *volleyball* in TbD-3D Dataset

Figure 6: **Qualitative comparisons of different methods.** The appearance reconstruction results for eight target time indexes are illustrated and the object is indicated by a red box in each image.

## F    ROBUSTNESS UNDER BACKGROUND SHIFTS

In the setting of the single image FMOs deblatting task Kotera et al. (2019), it is assumed that the camera is fixed with static backgrounds. In real-world scenarios, sometimes it is difficult to obtain backgrounds exactly matched to the blurred images. However, this situation would get easier when translation shifts is tolerable, which drives us to explore the performance of the methods under background shifts. To show the robustness of the methods under background shifts, for each sequence in test benchmarks, we shift the backgrounds in the direction sampled from 8 directions { left, right, upper, lower, left-upper, right-upper, left-lower, right-lower } uniformly. From the results in Table 7,

Table 6: Evaluation results with SfB.

| Prior model | Falling | | | TbD-3D | | | TbD | | |
|---|---|---|---|---|---|---|---|---|---|
| | TIoU | PSNR | SSIM | TIoU | PSNR | SSIM | TIoU | PSNR | SSIM |
| DeFMO | 0.650(0.001) | 24.98(0.10) | 0.732(0.003) | **0.864(0.000)** | 23.98(0.03) | 0.640(0.002) | 0.561(0.001) | 24.16(0.03) | 0.592(0.002) |
| LDINet | **0.658(0.012)** | **25.35(0.42)** | 0.730(0.004) | 0.843(0.003) | 24.17(0.10) | 0.651(0.001) | 0.601(0.000) | 24.13(0.06) | 0.609(0.001) |
| RCD | 0.641(0.010) | 25.32(0.06) | **0.745(0.002)** | 0.851(0.006) | **24.48(0.06)** | **0.659(0.001)** | **0.611(0.006)** | **24.43(0.20)** | **0.620(0.003)** |

Table 7: **Results of methods under background shifts.** We control the overlapping ratio between the shifted and original backgrounds to be 0.9.

| Model Name | Falling | | | TbD-3D | | | TbD | | |
|---|---|---|---|---|---|---|---|---|---|
| | TIoU | PSNR | SSIM | TIoU | PSNR | SSIM | TIoU | PSNR | SSIM |
| DeFMO | 0.482 | 24.59 | 0.706 | 0.713 | 24.70 | 0.645 | 0.452 | 24.42 | 0.560 |
| LDINet | 0.584 | 26.15 | 0.721 | 0.880 | 25.67 | 0.674 | 0.507 | 24.58 | 0.585 |
| RCD | **0.589** | **26.53** | **0.733** | **0.883** | **25.84** | **0.679** | **0.526** | **24.72** | **0.586** |

LDINet and RCD are more robust than DeFMO especially for the complex cases in the Falling Object dataset.

