# OpenReview forum: "LDINet:  Latent Decomposition-Interpolation for Single Image Fast-moving Objects Deblatting"
_ICLR.cc/2025/Conference — ICLR 2025 Conference Withdrawn Submission_

### Official Review · Reviewer_UZC1 · 2024-10-28

**Soundness:** 2
**Presentation:** 2
**Contribution:** 2
**Rating:** 3
**Confidence:** 4

**Summary:**

The authors proposed a method for deblurring and matting of fast moving objects. Motivated by the previous work, DeFMO, the authors modified it by first decomposing the latent into different parts and then interpolating them for decoding.  Such a decomposition followed by interpolation approach is designed with affine transformation subnetworks, and its effectiveness is justified by several experiments. The loss terms are appropriately defined for training the network.

**Strengths:**

1. The proposed method is motivated by the problem of the existing work, DeFMO. It is persuasive to construct varying latent representations for different time indices, rather than using a share latent for decoding.
2. The proposed module, DIM, and loss functions are well designed, and their effectiveness is demonstrated by sufficient ablation studies.

**Weaknesses:**

1. The proposed method is strongly motivated by DeFMO, which is outdated. The authors only refined the features between the encoder and decoder of DeFMO, which is incremental.

2. The motivation of the decomposition first and interpolation later does not have theoretical support. Since such two parts can be readily integrated, more detailed support is expected for better understanding.

3. The task of deblatting of FMOs is somewhat restricted. The problem setting of requiring the background image is not always practical, and complicated background environments are not tested. Thus, this reviewer consider that some modifications from DeFMO have limited technical contribution.

**Questions:**

1. The method decomposes a shared feature V. Why not extract decomposed features directly by an encoder?
2. More detailed analysis and visualizations on the AffNet will be helpful for better understanding.
3. Since the network is trained on synthetic datasets, can't we perform intermediate supervision to P, Q, and A?

---

### Official Review · Reviewer_7T87 · 2024-10-31

**Soundness:** 3
**Presentation:** 3
**Contribution:** 3
**Rating:** 8
**Confidence:** 2

**Summary:**

This paper addresses the problem of image deblatting of fast moving objects. This work improves up on the previous attempt in the domain by changing the way the latent embedding is decoded into multiple sharp frames. To this end, authors propose to bring in the possibility to assume linear motion over small time interval and then modeling the changes from one part of the time frame to another using affine motion. Such additional constraints and regulations helps to recover better mask as well as sharp objects as highlighted through various examples.

**Strengths:**

- The core idea to constraint the deep representations with the knowledge and approximations possible for motion is intuitive and also form a  significant contribution
- Proposed method achieves state-of-the-art results on the targeted task as demonstrated via multiple public datasets

**Weaknesses:**

- Imapct of each changes in the ablation study with visual examples and corresponding explanations on the different effects brought in by ideas in the proposed method would have been insightful for the readers.

**Questions:**

Please address my comments under the weaknesses

---

### Official Review · Reviewer_GcrZ · 2024-11-04

**Soundness:** 2
**Presentation:** 3
**Contribution:** 2
**Rating:** 5
**Confidence:** 4

**Summary:**

The paper introduces LDINet (Latent Decomposition-Interpolation Network), a novel method for deblurring fast-moving objects (FMOs) in single images. LDINet decomposes the motion blur into several shorter blur parts and uses an efficient Decomposition-Interpolation Module (DIM) to break down the feature maps into discrete time-indexed parts. The method then interpolates the target latent frames using affine transformations and renders sharp and clear images with a decoder. Additionally, the paper presents a Refining Conditional Deblurring (RCD) approach to further enhance the fidelity of textures and the accuracy of masks. Extensive experiments demonstrate that LDINet outperforms existing methods in deblurring FMOs.

**Strengths:**

- Effective Decomposition: The Decomposition-Interpolation Module (DIM) effectively breaks down the feature maps into discrete time-indexed parts, addressing the challenge of motion blur.
- Affine Transformations: The use of affine transformations for latent frame interpolation provides high-quality results, particularly in areas where linear interpolation fails.
- Refining Framework: The RCD approach enhances the results of LDINet, further improving the fidelity of textures and the accuracy of masks.

**Weaknesses:**

- Limited Dataset Diversity and Lack of Real-World Applications: The experiments are primarily conducted on the Falling Objects, TbD, and TbD-3D datasets, which may not fully represent the diversity of real-world video content. For example, can the authors provide results from different real-world sports scenarios such as basketball, soccer, table tennis, badminton, etc.?
- Computational Efficiency: The computational efficiency of the method, especially in comparison to existing methods, is not thoroughly analyzed.

**Questions:**

- Computational Efficiency: How does the computational efficiency of LDINet compare to existing methods, particularly in terms of inference time and memory usage?
- Resource Requirements: Could you provide more detailed information about the hardware and software requirements for running LDINet, especially for large-scale deployments?
- Is there any limitation to your method? For example, scenes that can't be handled or discontinuous results?

---

### Note · Authors · 2024-11-28

I have read and agree with the venue's withdrawal policy on behalf of myself and my co-authors.